

# Disturbance history alters the development of the HPA axis in altricial nestling birds

Ahmad Barati[1], Ondi L. Crino[2,3,4], Paul G. McDonald[5] and
Katherine L. Buchanan[6]

[1] Avian Behavioural Ecology Laboratory, Zoology, University of New England, Armidale, NSW, Australia
[2] School of Science and Engineering, Flinders University, Bedford Park, SA, Australia
[3] Division of Ecology and Evolution, Research School of Biology, The Australian National University, Canberra, ACT, Australia
[4] Faculty of Science, Engineering and the Built Environment, Deakin University, Geelong, VIC, Australia
[5] University of New England, Avian Behavioural Ecology Laboratory, Zoology, Armidale, NSW, Australia
[6] Deakin Centre of Integrative Ecology, School of Life and Environmental Sciences, Faculty of Science, Engineering and the Built Environment, Deakin University, Geelong, Vic, Australia

Corresponding author
Ahmad Barati,
barati728@yahoo.com

## ABSTRACT

Glucocorticoid (GC) hormones regulate the vertebrate stress response and are secreted by the hypothalamic–pituitary–adrenal (HPA) axis. Acute elevation of GCs is thought be adaptive because it promotes physiological and behavioural changes that allow animals to cope with disturbances. In contrast, chronic elevation of GCs is associated with reduced body condition, immune function, reproductive success, and survival. In adult birds, the effects of chronic stress have been well documented, including human-related disturbances. In contrast, the effects of chronic stress on nestlings have rarely been addressed. This is of interest, as many ecological or monitoring studies of wild birds require nestlings to be regularly handled. However, the consequences of repeated handling of nestlings on HPA axis function and body mass in wild birds remain poorly quantified. We examined whether daily exposure to handling stress increased corticosterone (the dominant avian glucocorticoid) secretion and reduced pre-fledging body mass relative to undisturbed control nestlings of the noisy miner (*Manorina melanocephala*), a native Australian passerine bird. Daily handling resulted in an elevated baseline and attenuated stress-induced corticosterone levels in disturbed 14-day nestlings, in comparison with control nestlings handled for the first time. Despite this, disturbed and control nestlings fledged at a similar body mass. Baseline and stress-induced corticosterone increased with nestling age but remained independent of nestling sex and hatching order. Our findings are some of the first to suggest that chronic handling stress causes physiological alterations to the development of the HPA axis in nestling birds, and our data suggest that researchers should minimise or account for handling stress in their experiments. These data also raise the possibility that other chronic stressors may have long term physiological consequences for the development of the HPA axis in nestling birds.

# INTRODUCTION

Vertebrates respond to a broad range of adverse conditions and events in their environment through the activation of the hypothalamic–pituitary–adrenal (HPA) axis (*Romero & Wingfield, 2016*; *Scanes & Dridi, 2021*). The HPA axis modulates both physiological and behavioural responses, allowing animals to cope with a range of both predictable and unpredictable stressors (*Wingfield, Vleck & Moore, 1992*; *Sockman & Schwabl, 2001*; *Romero, 2002*). The HPA axis regulates production of glucocorticoid (GCs) hormones, which mediate the balance between metabolic demands and energy stores *via* gluconeogenesis, including the energy demands associated with growth and development. Glucocorticoids play an important role in immune regulation, with prolonged elevations of GCs also suppressing immune function and increase anti-inflammatory proteins (*Spencer, Kaman & Dhabhar, 2011*; *Vleck et al., 2000*; *Räberg et al., 1998*), as well as having consequences for behaviour and memory formation (*Wingfield et al., 1998*; *Wingfield & Kitaysky, 2002*; *Jones et al., 2016*). In the short term, elevated GCs are thought to enhance survival and fitness, at the cost of reproduction (*Wingfield et al., 1998*; *Sapolsky, Romero & Munck, 2000*). Long term or chronic elevations of GCs are associated with reduced body condition and are presumed to have negative effects on fitness (*Cyr & Romero, 2007*).

Birds show a gradual development of the HPA axis in early life (*Wada et al., 2009*). The rate of development of the HPA axis varies with life history strategy, with precocial species exhibiting fuller development of the HPA axis prior to hatching, in comparison to altricial species (*Zimmer & Spencer, 2014*; *Berg et al., 2019*; *Bebus, Jones & Anderson, 2020*). In altricial nestlings, endocrine responses to stressors are thought to be adaptive, as they facilitate the direction of resources to the development and growth of tissues (*Wada, Hahn & Breuner, 2007*; *Bebus, Jones & Anderson, 2020*; *Jones, Nguyen & Duval, 2021*), enhancing growth rate and enabling fledglings to more quickly avoid stressful conditions. In contrast, the delayed development of the HPA response in altricial species has been suggested to functionally serve to protect young nestlings from the adverse impact of chronic stress, when they are unable to actively avoid it (*Blas et al., 2006*).

Apart from natural stressors such as thermal stress and reduced food availability and predator presence, birds can also be exposed to stress due to human-related activities and disturbances (*Frid & Dill, 2002*; *Crino et al., 2011*; *Blickley, Blackwood & Patricelli, 2012*; *Fokidis, Orchinik & Deviche, 2009*). Avian responses to human disturbance are thought to be largely analogous to responses triggered by natural predation risk, resulting in similar behavioural and physiological reactions (*Frid & Dill, 2002*). For example, birds provision nestlings less and produce louder alarm calls in response to human presence and activity near their nests (*Müller et al., 2006*). Other types of activities, such as live-trapping and handling, can also increase corticosterone (the main avian GC) (*Lynn & Porter, 2008*).

Although the short-term effects of handling on stress responses have been well studied, far less is known about the sustained effects of repeated handling on HPA axis function.

Most of the studies that have examined handling effects, assess adult birds (*e.g.*, *Schwabl, Bairlein & Gwinner, 1991*; *Rich & Romero, 2005*; *Creel et al., 2013*) and the impact of human-related disturbances on developing birds is not fully understood with only limited studies (*e.g.*, *Sagar et al., 2019*; *Brewer, O'Reilly & Buck, 2008*; *Bebus, Jones & Anderson, 2020*). The handling of avian nestlings is a common ecological monitoring practice for marking or measurement to quantify growth, development, and reproductive success. All these activities can inadvertently cause both behavioural and physiological responses, including activation of the HPA axis and increased production of corticosterone (*Wingfield & Romero, 2001*). Understanding the long-term effects of handling on developing birds is especially important, because exposure to stress during development can result in sustained changes in HPA axis function, plausibly through a range of different mechanisms with long lasting impacts into adulthood (*Romero & Wingfield, 2016*) and even intergenerational effects (*Kraft, Crino & Buchanan, 2021*; *Spencer, Evans & Monaghan, 2009*; *Zimmer et al., 2017*). Few studies have examined the impact of handling stress on the development of the HPA axis on wild birds and those that have, have mainly focused on precocial and semi-precocial species (*e.g.*, *Watson et al., 2016*). There are indeed limited studies addressing the impact of handling stress on non-precocial/altricial species in the wild (*e.g.*, *Adams et al., 2008*; *Quillfeldt et al., 2009*) and we are unaware of any such studies on passerine species, which are commonly handled for research purposes. As the impact of handling on growth, development and HPA programming seem likely to differ between species there is a need to quantify how handling impacts on the development of the HPA axis in altricial nestlings.

The objectives of this study were therefore to investigate how repeated handling during the nestling period impacts the nestling growth and development as well as the endocrine stress response of an altricial bird, the noisy miner (*Manorina melanocephala*). The noisy miner is a native Australian honeyeater from the Meliphagidae family and is endemic to wooded country in south-eastern Australia. The noisy miner is an obligate cooperative breeder that forms large colonies with complex internal social structures consisting of breeding pairs and helpers (*Dow & Whitmore, 1990*). Nestlings in this species are altricial, fledging at approximately 15 days post-hatch. As fledglings they are heavily dependent on care and provisioning provided by breeders and helpers alike until they become fully independent at around 3 months old (*Barati et al., 2018*; *Barati, Andrew & McDonald, 2021*). We examined the impact of human disturbance on corticosterone responses and the body size of miner nestlings prior to fledging. Repeated handling leads to attenuated stress response in poultry (*Hemsworth et al., 1994*) and there is some evidence for a similar effect in Amazon parrots (*Collette et al., 2000*), suggesting that repeated exposure to a stressful event dampens the HPA axis response during long term exposure. We therefore predicted that 1) repeatedly handled and disturbed nestlings would have higher levels of baseline corticosterone and lower levels of peak corticosterone late in nestling period, and 2) repeated nestling handling would be associated with a decrease in age-specific body mass compared to control groups, due to the suppressive effects of elevated levels of corticosterone on growth rates in nestling birds (*e.g.*, *Spencer et al., 2003*;
*Crino, Driscoll & Breuner, 2014*; *Kraft et al., 2019*), thought to be mediated through changes in begging behaviour (*Kitaysky, Wingfield & Piatt, 2001*; *Loiseau et al., 2008*).

## METHODS

### Study populations and general field methods

This study was undertaken at two focal colonies of the noisy miner: Newholme Field Research Station, a working rural property located 12 km north of Armidale, NSW (30°25′ 24″S; 151°38′84 38″E) and Dumaresq Dam Public Reserve, 12 km north-west of Armidale (30°30′S, 151°40′E) Australia. This study was carried out in accordance with the approved (Protocol AEC13-142) guidelines and regulations of University of New England, Animal Ethics Committee. The methods used in the study followed the *Australian code for the care and use of animals for scientific purposes* and we aimed to use a minimum number of animals and the to minimise disturbance to individuals subject to the study. Fieldwork was conducted during the 2014–2015 breeding seasons from mid-August to late November each year. To locate nests, study sites were surveyed every 2–3 days from mid-August. Nests were marked using a plastic tag attached to a nearby tree and nests contents were checked with direct observation (using a ladder) or by a mirror attached to the end of a 10-m pole to determine hatching dates (*Barati, Etezadifar & McDonald, 2016*; *Barati & McDonald, 2017*). From the expected hatching date onwards (around 14 days after the eggs were laid), nests were visited daily to record actual hatch date and monitor clutch progress. To identify nestlings within broods, we marked each nestling on their leg with a non-toxic marker until around 7–8 days of age, when nestlings were ringed with a metal leg ring issued by the Australian Bird and Bat Banding Scheme. Before fledging, nestlings were banded with a combination of three colour bands, of which one colour band was fitted with a small passive integrated transponder (PIT) tag (Nano Tag, Biomark, Boise, ID, USA). Broods were placed into either treatment ($N = 15$) or control ($N = 9$) broods randomly, with these groups subject to different handling and disturbance regimes.

*a) Treatment broods* were subject to handling and measurements of nestlings daily. Measurements were undertaken every day between 14:00 and 18:00 h. On each occasion, broods were taken from the nest and linear growth measurements were conducted within 1 min of removal. This involved weighing nestlings on a top-pan balance (Mini-Table-Top One Balance Vaporize, Australia; accuracy 0.01 g) and measuring head to bill and tarsus length to the nearest 1 mm using callipers. When nestlings where 8, 11, and 14 days old, respectively, prior to the other growth measurements being taken and immediately after first handling the chicks, we also collected a small blood sample (~150 μL) from the brachial vein using sterile needles (27 g) and heparinised microcapillary tubes. b) *Control broods* were monitored from a distance (approximately 20 m) using binoculars to confirm that nests were active, but nestlings were not handled directly during monitoring. Instead, control broods were handled only once at 14 days post-hatch, when blood samples and measurements were recorded as per the treatment broods at that same age. All nestlings at a given nest were immediately returned to the nest when measurement and sampling of the last nestling was completed.

## Sample processing

Blood sampling and handling of birds were carried out in accordance with the approved (Protocol AEC13–142) guidelines and regulations of University of New England, Animal Ethics Committee. Initial blood samples were collected within 3 min of first handling a given nestling, but we also recorded the exact time of sampling (seconds) from the initial handling of the first nestling of a brood until the blood collection of each nestling had been completed.

As the aim of the study was to measure both baseline corticosterone levels and stress-induced increases, it was critical to note the exact time of the onset of increasing the corticosterone after first disturbance (*Romero & Reed, 2005*; *Small et al., 2017*). Thus, we recorded the exact time from the first touch of the first nestling until sampling was completed for each nestling. Previous sampling analyses of adult noisy miners suggested stress-induced corticosterone levels reach a peak around 30 min post-disturbance (Barati et al., 2018, personal observations). Therefore, a second sample was collected from nestlings 30 min from the onset of handling. Between the first and 30 min sample collection, nestlings were kept in an opaque cloth bag, placed in the shade. In broods with more than two nestlings ($N = 8$), only two nestlings were selected randomly at the first blood sampling for corticosterone assay to minimise handling time. Average handling time was approximately 30 min at each nest as the final sample collection was quickly completed and birds returned to the nest. Noisy miners are very tolerant of disturbance at the nest, and resumed feeding only a few minutes after we had left the nest area in most cases (Barati et al., 2018, personal observations). Thus, the impact of nest disturbance is unlikely to confound the results related to chronic HPA activation. In control broods, one random nestling was selected from each brood and blood samples were collected in the same manner when nestlings were 14 days old. Fourteen days is close to the age for fledgling in noisy miner, which is usually fledge at 15 days post-hatch in our study populations (*Barati, Andrew & McDonald, 2021*; *Barati & McDonald, 2017*). In total, blood samples from 45 nestlings belonging to 24 different broods were collected. Blood samples were stored in heparinised capillary tubes on ice, before being transferred to the lab where they were centrifuged at 20,800 g for 6 min. Plasma was then separated from the blood cells and stored in 1.5 ml Eppendorph tubes at −20 °C until corticosterone levels were assayed. The remaining blood cells from capillary tubes were then used as a source of DNA to enable each nestling to be sexed molecularly based on sexing polymerase chain reactions (PCRs) (*Griffiths et al., 1998*; see *Barati et al., 2018*).

## Corticosterone assay

Corticosterone levels were quantified with Enzyme Immunoassay (EIA) kits (Cat No. ADI 900-097; Enzo Life Sciences, Farmingdale, NY, USA). Samples were spiked with one picogram of tritiated corticosterone prior to steroid extraction to determine individual sample recovery percentage. We extracted corticosterone from raw plasma using a double wash of dichloromethane. Samples were then dried under nitrogen gas and reconstituted in buffer solution (1:30 ratio). The average sample recovery was 72% (range 67–80%) and we adjusted hormone values according to individual recovery values. We ran our

reconstituted samples at half volume against a six-point standard curve, ranging from 20,000 to 15.53 pg/ml (as per *Crino et al., 2017*). An external standard of 500 pg/ml was run on every plate and used to calculate inter-plate variation. All samples and standards were run in triplicate. Plates were read on a FLUOstar Omega microplate reader at 405 nm corrected at 595 nm. Intra- and inter-plate variation was 3.95% and 5.16% respectively.

## Statistical analysis

Statistical analyses were conducted in the R environment (*R Core Team, 2021*). All modellings were performed in *lme4* package (*Bates et al., 2014*). First, to investigate if baseline corticosterone varied within 3 min of sampling, we used a generalized linear mixed effect model (GLMM) model with the timing of sampling (seconds) as a predictor variable and age of nestling and brood size as random effects. Response variable was the baseline level of corticosterone with a Gaussian distribution (link= identity). To examine how baseline and stress-induced level of corticosterone vary with the nestling age in broods subject to multiple sampling at different ages, we performed a (GLMM) with nestling age (three levels) as a fixed effect and nest identity, hatching order and brood size as random effects. Response variables were either baseline or stress-induced corticosterone level tested with a Gaussian distribution (link= identity). In both tests above, the significance of the fixed effect was examined with a likelihood ratio test (LRT) in which the model with fixed effect was compared to intercept only model. If there was a significant effect, we performed a Tukey *post-hoc* test to examine within-subject differences using the package *multcomp* in R (*Hothorn, Bretz & Westfall, 2008*). We used same approach for testing whether body mass and corticosterone differed between sexes. Body mass, baseline and stress-induced corticosterone were response variables in the models and nestling sex a fixed effect, while finally nest identity and hatching order were included as random effects. We used a Pearson's product-moment test to examine correlation between pre-fledging body mass and corticosterone levels. To examine variation of corticosterone with sampling time (*i.e.*, baseline stress-induced corticosterone), we fitted generalized linear models (GLMs) with corticosterone levels as the dependent variable and time between disturbance and sampling as the predicting factor. Effect sizes and significance levels are reported for each test. Finally, to test if handing disturbance influences baseline corticosterone, stress-induced corticosterone, pre-fledging body mass and tarsus length, we fitted GLMMs to corticosterone levels and body measurements as a response variable with a Gaussian distribution (link= identity). Nest identity, nestling sex, brood size and hatching order were included as random effects to account for multiple sampling at the same nest and to control for the effect of hatching order and sex. Significance of the fixed effect was examined with an LRT in which the model with fixed effect was compared to intercept only model. In the GLMM models, the significance of each factor was assessed based on the estimated effects sizes and 95% CIs (*Nakagawa & Cuthill, 2007*).

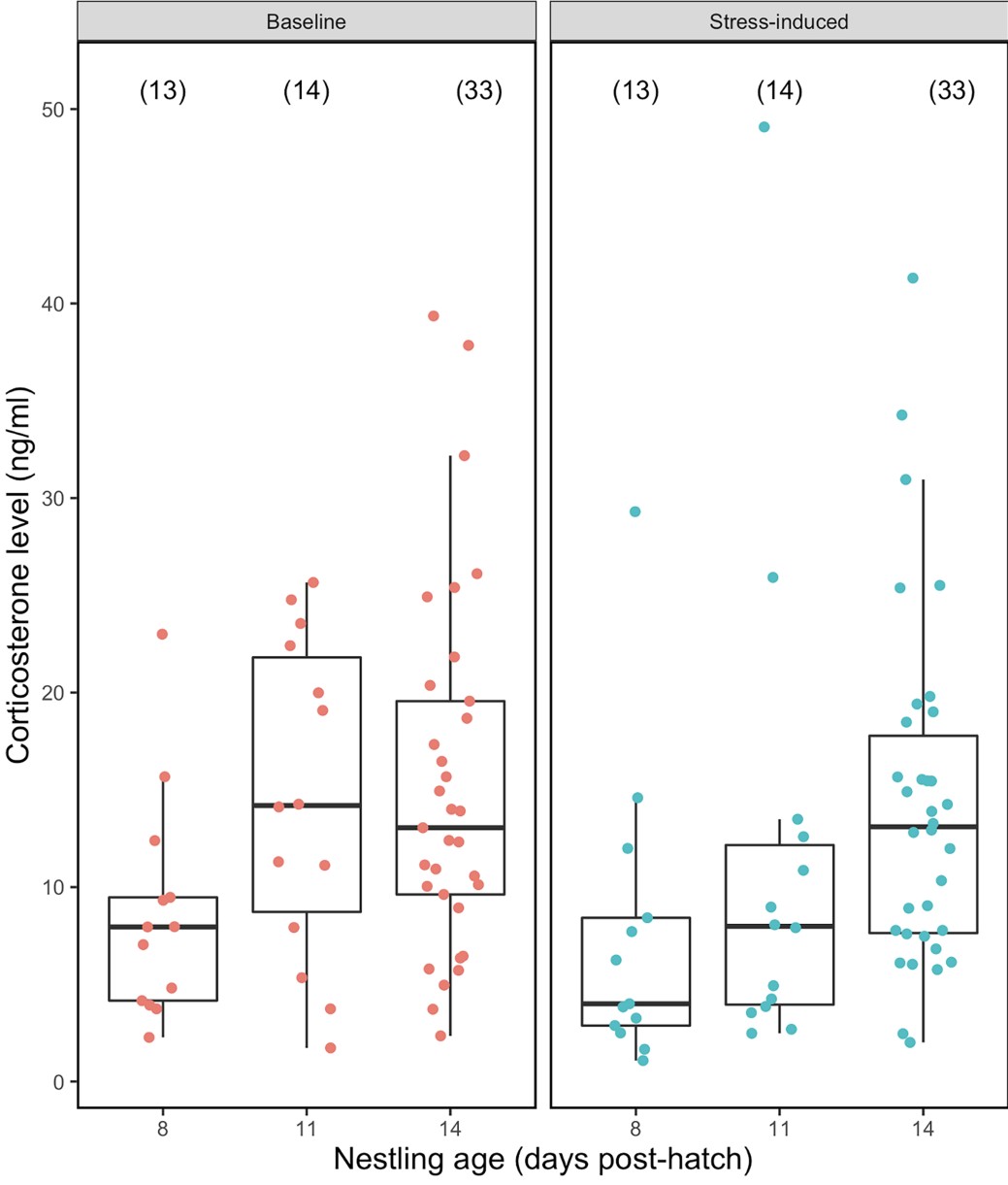

**Figure 1 Box plot showing the baseline (time 0 min post capture) and stress-induced (time 30 min post capture) corticosterone levels in disturbed nestlings according to their age.** Lower and upper box boundaries represent 25th and 75th percentiles, respectively, line inside box is median, lower and upper error lines of the 10th and 90th percentiles, respectively. Sample sizes for each age category are given in brackets.

## RESULTS

### How does corticosterone level vary with age, sex and body mass?

Sampling time of nestlings from the initial disturbance varied from 59 to 180 s with average time being 113.3 ± 12.3 (mean ± SE) seconds and the sampling time did not affect baseline corticosterone levels (GLMM, $\chi^2_1 = 0.19$, $p = 0.65$). Disturbed birds were

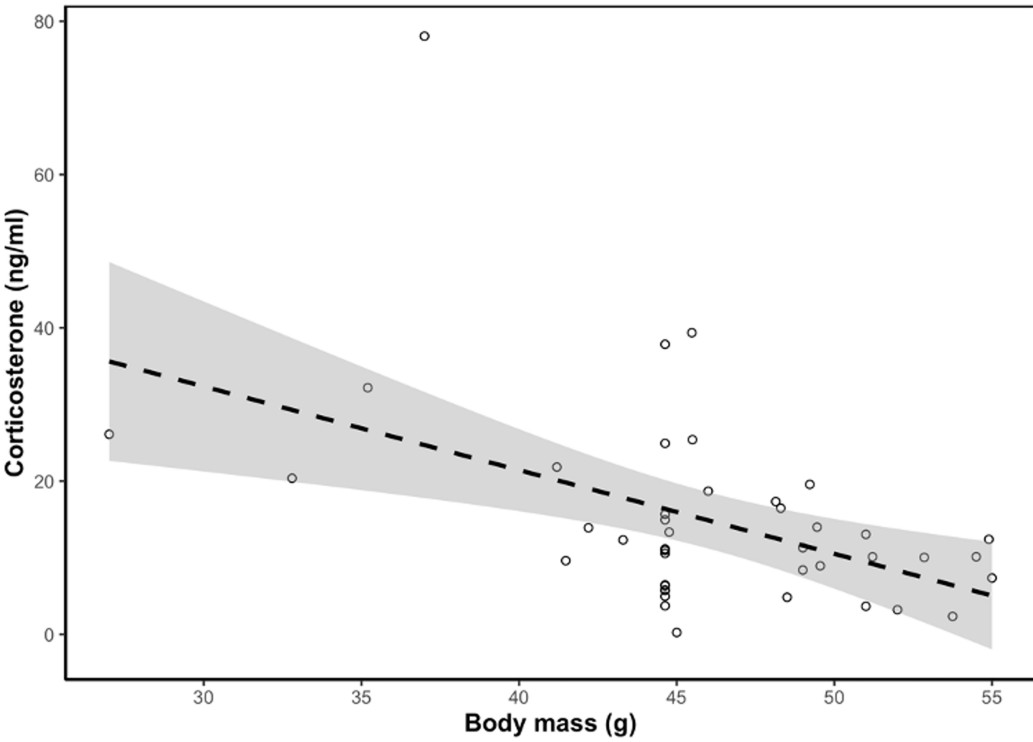

**Figure 2 Variations of baseline corticosterone with body mass (grams) for day 14 nestling noisy miners.** Dashed line shows the best linear fitted line and shaded areas display 95% confidence intervals around the fitted line.

subject to three age-specific sampling events at the ages of 8, 11 and 14 days post-hatch. Both baseline and stress-induced corticosterone increased as nestlings became older (Fig. 1). These trends were found to be statistically significant for baseline corticosterone (GLMM, $\chi^2_2 = 7.04$, $p = 0.02$). *Post-hoc* Tukey comparisons suggested that corticosterone levels at day 8 was significantly lower than corticosterone levels at day 14 ($\beta = -0.71$, se = 0.24, $p = 0.01$). Similar patterns were observed for stress-induced corticosterone levels that significantly increased with the age of nestlings (GLMM, $\chi^2_2 = 5.59$, $p = 0.05$; Fig. 1). *Post-hoc* Tukey tests identified corticosterone levels at day 8 were significantly lower than corticosterone levels at day 14 ($\beta = -0.69$, se = 0.28, $p = 0.04$).

As expected, due to the sexual size dimorphism in this species, male nestlings fledged with a higher body mass compared to female nestlings (GLMM, $\chi^2_2 = 6.19$, $p = 0.01$, $N = 32$ female and $N = 38$ male nestlings). However, neither baseline or stress-induced corticosterone levels varied significantly between sexes, after controlling for hatching order and handling treatment (baseline: GLMM, $\chi^2_2 = 0.39$, $p = 0.52$; Stress-induced: GLMM, $\chi^2_2 = 1.26$, $p = 0.26$; $N = 32$ female and $N = 38$ male nestlings).

Combined across all samples, at day 14 post-hatch, baseline corticosterone was significantly negatively correlated with nestling body mass (Pearson's product-moment correlation, t = $-3.35$, df = 41, $p = 0.001$, $R^2 = -0.46$) (Fig. 2). However, stress-induced corticosterone remained independent of nestling body mass, at this age (Pearson's product-moment correlation, t = 1.33, df = 41, $p = 0.19$).

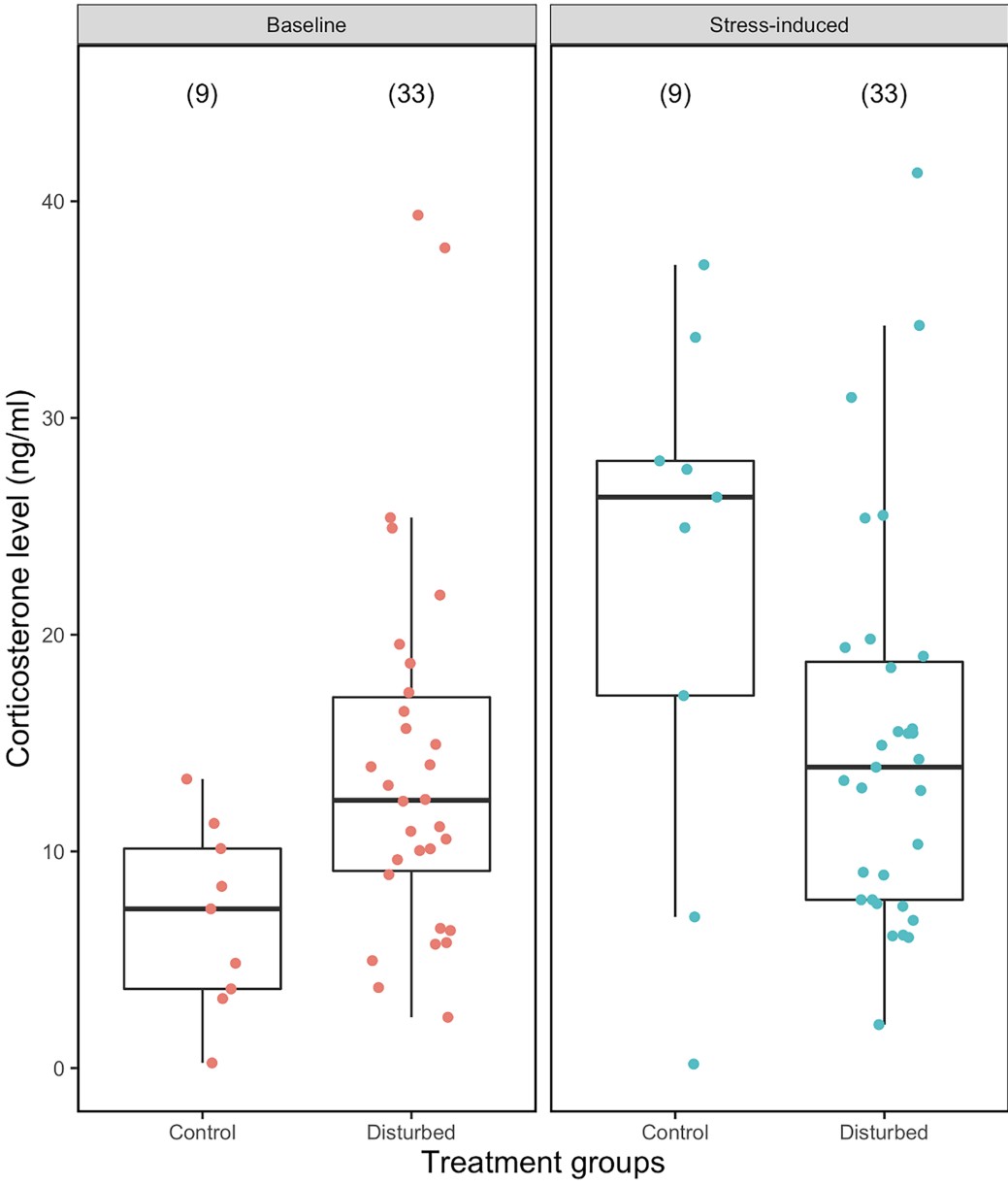

**Figure 3** Box plots showing the baseline and stress-induced (time 30 min post capture) corticosterone levels in disturbed and control nestling noisy miners at age 14 days post hatch. Lower and upper box boundaries represent 25th and 75th percentiles, respectively, line inside box is median, lower and upper error lines 10th and 90th percentiles, respectively. Sample sizes for each treatment group are given in brackets.

## Variation of corticosterone with sampling time in control and disturbed nestlings

For control birds, corticosterone levels increased significantly, as expected 30 min after initial disturbance (GLMM, $\beta$ =15.51, se = 4.29, $p$ = 0.002; baseline: 6.93 ± 1.42 ng/ml $vs.$ stress-induced: 22.45 ± 4.05 ng/ml, $N$ = 9, all control birds age 14 days). In disturbed birds however, there was no such significant difference between the baseline and stress-induced

corticosterone at day 14 post-hatch (GLMM, $\beta = -2.57$, se = 1.97, $p = 0.19$; $N = 33$; Fig. 3), demonstrating attenuation of the stress response by this age. Similarly, baseline and stress-induced corticosterone of handled birds did not vary at ages 8 and 11 days post-hatch (Day 8 post-hatch: GLM, $\beta = -1.09$, se = 2.66, $p = 0.68$; Day 11 post-hatch: GLMM, $\beta = -3.311$, se = 3.98, $p = 0.41$, $N = 14$; $N = 13$, Fig. 1).

### Does handling disturbance influence baseline and stress-induced corticosterone and pre-fledging body mass?

After controlling for hatching order and sex, baseline corticosterone levels were higher in handled nestlings compared to control nestlings being sampled for the first time (GLMM, $\chi^2_1 = 4.26$, $p = 0.03$, $\beta = 10.03$, se = 4.8; Fig. 3). In contrast, stress-induced corticosterone levels were significantly lower in handled nestlings compared to control nestlings (GLMM, $\chi^2_1 = 5.02$, $p = 0.02$, $\beta = -8.19$, se = 3.62; Fig. 3). In control nestlings there was a rapid increase of corticosterone 30 min after first sampling, but in the disturbed group, baseline and stress-induced corticosterone remained similar at all ages prior to fledging (Figs. 1 & 3). Finally, we tested if handling stress impacted nestling body mass and tarsus length before fledging. All nestlings subject to measurements and sampling survived and successfully fledged. After controlling for hatch order and sex, frequent handling of nestlings did not result in a difference in pre-fledging body mass (GLMM, $\beta = 4.7$, se = 2.4, $p = 0.06$) and tarsus length of nestlings (GLMM, $\beta = 1.2$, se = 0.71, $p = 0.25$).

## DISCUSSION

Early developmental experiences can have profound impacts on adult physiological responses, including altered HPA axis function, resulting in changes in circulating glucocorticoid levels and receptor function (Buchanan et al., 2003; Crino & Breuner, 2015). Such change may even impact across generations (Zimmer et al., 2017; Kraft, Crino & Buchanan, 2021). Here, we show that chronic stress through repeated handling and blood sampling affects HPA axis function in nestlings of a common altricial bird species. We found that nestlings that were handled daily had both elevated baseline and attenuated stress-induced corticosterone levels. These results suggest that chronic handling can cause physiological alterations to the development of the HPA axis in nestling birds. Additional research is needed to determine both the immediate and long-term consequences for post-fledgling development, survival, and fitness.

The effects of repeated handling stress on precocial birds have been well documented. However, precocial and semi-precocial species hatch at a much more advanced state of development compared to altricial birds making direct comparisons between the two development strategies unreliable (i.e., Fiske, Gannon & Newman, 2013, Zimmer et al., 2017). The avian stress response is hypothesised to develop at a rate which reflects its adaptive role in mediating responses, with altricial species showing delayed development (Blas et al., 2006; Wada et al., 2009; Romero & Wingfield, 2016). Watson et al. (2016) detected no effect of repeated handling of semi-precocial European storm petrel (Hydrobates pelagicus) nestlings on baseline corticosterone levels or body size of nestlings. It is worth noting that Watson et al. (2016) handled nestlings at different age intervals and

not daily, which can make it difficult to directly compare their results with our current study, with regular daily handling over the nestling period.

## Age-related changes in the HPA axis

In nestling noisy miners, both baseline and stress-induced corticosterone levels increased with nestling age. Nestling responses to stressful conditions depended on factors such as their nutritional dependency, as well as key physiological conditions including capacity to thermoregulate and locomote (*Starck & Ricklefs, 1998*). Thus, a stress hypo-responsive period, which occurs in many altricial mammal species (*Romero & Wingfield, 2016*), is also predicted to occur in altricial avian nestlings, in order to mitigate the detrimental effects of a stress response during early dependency on paternal care (*Blas et al., 2006*; *Wada et al., 2009*). There is good evidence to support the hypothesis that delayed responsiveness is an adaptive developmental strategy from a number of passerine species (*Romero & Wingfield, 2016*). For example, in the altricial zebra finch (*Taeniopygia guttata*) nestlings, corticosterone levels show substantial age-related individual variation and baseline corticosterone predicted fledging body mass (*Wada et al., 2009*), although evidence from precocial species is more mixed (*Romero & Wingfield, 2016*).

When compared between control and disturbed birds, repeated handling caused an elevated baseline corticosterone in disturbed birds, whereas stress-induced corticosterone decreased in disturbed birds. Together, our findings suggest that repeated exposure to handling stress elevates baseline corticosterone levels, but attenuates stress-induced response in these altricial nestlings. As the noisy miner has an altricial developmental pathway, the endocrine response to handling stress is in line with studies documenting a stress hypo-responsive period. For example, in semi-precocial nestling European storm petrels, handling intensity did not have any effect on baseline corticosterone and nestlings appear to be unresponsive to handling effects from the investigators (*Watson et al., 2016*). Similarly, young birds did not show a HPA axis-related response (*i.e.*, variations in the baseline or stress-induced corticosterone levels) to handling disturbance in Leach's storm-petrel (*Oceanodroma leucorhoa*) (*Fiske, Gannon & Newman, 2013*) and mottled petrel (*Pterodroma inexpectata*) before the age of 6 weeks (*Sagar et al., 2019*). It has been suggested that the absence of hormonal response in some precocial species could be related to how birds perceive handling as stressful event and the possibility that the function of HPA axis is not strongly associated with stressful stimuli early in the life (*Watson et al., 2016*). There are however, fewer studies addressing any possible stress hypo-responsive period in precocial species (*Romero & Wingfield, 2016*). Therefore, when assessing the impacts of handling disturbance, it is also important to note that altricial and non-altricial species may show species-specific endocrine responses, which generally reflect their life history.

For altricial nestlings, their responses are thought be diverse within different taxonomic groups (*Adams et al., 2008*; *Quillfeldt et al., 2009*). For example, higher levels of baseline corticosterone observed in disturbed nestling noisy miners is inconsistent with responses reported for nestling American kestrels (*Falco sparverius*) (*Dufty, 2008*). In American kestrels, baseline corticosterone levels were similar between birds that had been disturbed

on both a single and multiple events. However, like noisy miner nestlings, nestling American kestrels that were regularly disturbed had significantly lower plasma corticosterone levels than birds blood sampled only once after the onset of capture-and-handling stress (*Dufty, 2008*). Future studies that test the impact of repeated handling on nestlings across species will all us to draw more robust conclusions on the impact of regular handling on HPA axis development in nestlings.

## Impact of handling disturbance on pre-fledging body mass

In noisy miner nestlings, baseline corticosterone level decreased as body mass increased in nestlings at 14 days of age. In contrast, we did not find evidence that stress-induced corticosterone level is associated with body mass in nestling noisy miners. The relationship between body condition and baseline corticosterone is rather inconsistent in altricial nestlings, with some studies reporting a positive correlation between body condition and the corticosterone level (*Love, Bird & Shutt, 2002*), while others have found the opposite relationship (*Nunez-de la Mora, Drummond & Wingfield, 1996*; *Sockman & Schwabl, 2001*). In altricial zebra finch nestlings, experimentally elevated corticosterone levels have been shown to negatively impact body condition (*Spencer, Evans & Monaghan, 2009*; *Crino, Driscoll & Breuner, 2014*).

In contrast to our prediction, nestlings subject to daily handling showed no immediate reduction in body mass gain. The lack of a difference between pre-fledging body mass of handled and control nestlings could be due to presence and activities of a helper contingent. The noisy miner is a cooperatively breeding species (*Dow & Whitmore, 1990*; *Barati et al., 2018*), with up to 12 helpers contributing to the provisioning of nestlings (*Barati et al., 2018*), which may have masked the impact of handling in these nestling noisy miners. Elevated corticosterone can also alter nestling begging, which results in higher feeding rate (*Perez et al., 2016*), thus nestlings may not be disadvantaged due to a higher baseline corticosterone level.

Although most of the impact of stress could be species-specific in nestling birds (*Adams et al., 2008*), findings from this study confirm that even relatively short, repeated sampling disturbances can have physiological consequences for nestlings. Whilst it is possible that some of our results may be explained by the repeated blood sampling *per se*, rather than the handling, this is unlikely as the total blood volume removed was relatively small in comparison to total blood volume. Interestingly, despite altered HPA axis function, there was little obvious impact on the pre-fledging body mass of the nestlings, suggesting a level of endocrine reprogramming even with subtle impacts. We suggest that the longer-term impacts of repeated disturbance should be considered when designing research protocols that involve repeated disturbance to nestling birds.

## Potential mitigation strategies for reducing handling stress

Based on the results of our study, it is crucial to consider alternative methods that minimise disturbance to nestlings when planning research that involves a handling component.

First, minimising the duration and frequency of handling sessions could be an effective approach and employed at all times. Shortening handling times and reducing the number of handling events can lower the cumulative stress experienced by individuals. Is has been suggested that reducing handling frequency can significantly lower corticosterone responses in birds (*Voss, Shutler & Werner, 2010*). While we aimed to keep handling to a minimum in our study, future research could explore the use of remote monitoring technologies, such as video or automated tracking systems, which further reduce the need for direct physical interaction (*Rutz & Hays, 2009*). Second, employing stress-reducing handling techniques, such as ensuring that the process is performed by experienced personnel, might minimise stress. For instance, *Romero & Reed (2005)* suggest that habituation to handling by consistently using familiar methods can significantly lower stress responses. Quick and efficient sample collection, followed by promptly returning nestlings to their nest, is another strategy to minimise disturbance.

Additionally, habituation techniques, such as gradually familiarising birds with human presence before handling, may reduce stress responses over time. Although time-consuming, this approach has been shown to be effective in minimising the impact of repeated sampling, particularly in wild bird populations (*Cyr & Romero, 2007*). Finally, environmental modifications, such as using shaded areas during handling to mimic the nest's natural environment, reducing noise, or minimising movement around the nest, may also reduce external stressors. This could be particularly relevant in field studies where environmental factors play a significant role in stress responses (*Cilulko et al., 2013*).

## Broader taxonomic implications

Our findings demonstrate that chronic handling stress during early development attenuates the acute stress response in noisy miner nestlings, providing evidence of physiological impacts on HPA axis function during critical developmental stages. Similar findings in other bird and mammal species, suggest that chronic stress can influence development of HPA axis function across vertebrates (*Dickens, Delehanty & Romero, 2009*). While our data are specific to birds, the conserved nature of the HPA axis across vertebrates suggests that these results may have relevance for other taxonomic groups exposed to similar stressors, such as repeated capture or handling (*Romero & Wingfield, 2016*) or captive breeding programs, where juveniles may be regularly handled (*Morgan & Tromborg, 2007*). Future research should aim to determine whether these effects are consistent across other vertebrate groups, particularly mammals, reptiles, or amphibians, to understand the broader implications (*Sheriff et al., 2011*). By extending such studies, researchers can improve handling protocols in both ecological monitoring and captive management, mitigating chronic stress impacts and enhancing animal welfare.

## ACKNOWLEDGEMENTS

We are grateful to Farzaneh Etezadifar who helped during blood sampling and banding.

### Funding

The School of Environmental and Rural Sciences, UNE, provided financial support for this project. The project was also supported by the Ecological Society of Australia through Holsworth Wildlife Endowment Grant. There was no additional external funding received for this study. The funders had no role in study design, data collection and analysis, decision to publish, or preparation of the manuscript.

### Grant Disclosures

The following grant information was disclosed by the authors:
The School of Environmental and Rural Sciences, UNE.
Ecological Society of Australia through Holsworth Wildlife Endowment Grant.

### Competing Interests

The authors declare that they have no competing interests.

### Author Contributions

- Ahmad Barati conceived and designed the experiments, performed the experiments, analyzed the data, prepared figures and/or tables, authored or reviewed drafts of the article, and approved the final draft.
- Ondi L. Crino conceived and designed the experiments, performed the experiments, analyzed the data, authored or reviewed drafts of the article, and approved the final draft.
- Paul G. McDonald conceived and designed the experiments, authored or reviewed drafts of the article, and approved the final draft.
- Katherine L. Buchanan conceived and designed the experiments, authored or reviewed drafts of the article, and approved the final draft.

### Animal Ethics

The following information was supplied relating to ethical approvals (*i.e.*, approving body and any reference numbers):

This study was carried out in accordance with the approved (Protocol AEC13-142) guidelines and regulations of University of New England, Animal Ethics Committee.

### Data Availability

The data is available at Dryad: Barati, Ahmad; Crino, Ondi; McDonald, Paul; Buchanan, Katherine (2024). Disturbance history alters the development of the HPA axis in altricial nestling birds [Dataset]. Dryad. https://doi.org/10.5061/dryad.tx95x6b48.

### Supplemental Information

Supplemental information for this article can be found online at http://dx.doi.org/10.7717/peerj.18777#supplemental-information.

# PeerJ

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
