# Peer review of "Disturbance history alters the development of the HPA axis in altricial nestling birds"

_PeerJ, doi:10.7717/peerj.18777_

## Round 0.1 · original submission · Minor Revisions

Your paper has been now seen by two referees - they provided valuable comments on the content and communication of your results. I'm sure that taking them into account will improve your paper's quality.

·

Basic reporting

The manuscript titled "Disturbance history alters the development of the HPA axis in altricial nestling birds" explores the effects of chronic stress on nestling birds, specifically focusing on the hypothalamic-pituitary-adrenal (HPA) axis development in the Noisy Miner, an altricial bird species. The study aims to understand how daily handling, as a form of stress, influences corticosterone levels and pre-fledging body mass in comparison to control groups that were not subjected to daily handling.

The manuscript is well-written, and I am happy to have had the chance to review it. It provides a comprehensive introduction and background that successfully situates the study within the broader context of ecological and physiological research on birds. This will be an important contribution to understanding the physiological impacts of chronic stress on bird development.

Introduction:

The introduction is concise and clear and well organized. It provides excellent background information about glucocorticoid function and HPA axis development in nestling birds.

Line 62-63: You may want to consider adding the follow reference for the influence of GCs on memory. To the best of my knowledge, it is the only study that links HPA axis function and long-last memory in a bird species. To fully disclose, this is my own work.

Jones, Blake Carlton, Sara E. Bebus, Stephen M. Ferguson, Philip W. Bateman, and Stephan J. Schoech. "The glucocorticoid response in a free-living bird predicts whether long-lasting memories fade or strengthen with time." Animal behaviour 122 (2016): 157-168.

Experimental design

Methods:
For treatment broods please provide how long the average nest visit was for daily visits, from first disturbance to leaving the area. There is a potential confound between chronic activation of the HPA axis, and less feeding by parents due to nest disturbance. Estimates of provisioning rates between control and treatment nests before during and after nest visits could elucidate this possibility. In its absence, I think you should recognize this possibility, with perhaps any explanation for why you may think the possible confound is unlikely to explain the results.

Please provide the range of % recovery for the cort assay.

Line 247: I’m wondering why brood size, hatch order, and sex are included as random effects rather than fixed effects. Random effects are typically used when the levels of a variable are seen as a sample of a larger population of possible levels, like nest or brood id. Further, the partial pooling (using a shrinkage estimator) that is employed by a mixed model for random effects makes sense particularly for samples with poorly represented levels with a relatively large overall sample size, such as brood id, but not sex or hatch order (Gelman & Hill 2006). How many levels are there for brood order and brood size? Can you elaborate on your decision for include these variable as random effects rather than fixed ones?

Gelman, A. and Hill, J., 2006. Data analysis using regression and multilevel/hierarchical models. Cambridge university press.

Validity of the findings

Results:

Thank you for including individual data points on the boxplots.

I think it would be valuable to see the data visualized for comparing body mass between controls and treatment nests.

With the morphometric data you collected, can you see if structural size or feather development differs between control and treatment nestlings?

Are you able to test differences in fledgling timing between controls and treatments?

Conclusions:

I thought the conclusions were well discussed. I wonder if we know much about the possibility of how chronic handling stress during nestling stage might influence adult phenotype. Perhaps from any studies employing a different type of stressor during the nestling phase and tracking individuals longitudinally.

Can you provide any discussion on potential mitigation strategies for reducing handling stress in ecological studies?

Additional comments

I found some lines that need spaces.

52 — space need between “axis” and “(Romero…”
103 — space needed between “…2016)” and “and even…”
259 — space need between “11” and “and 14…”

·

Basic reporting

The study provides a detailed description of the methods. The data collection procedures appear to be rigorous.

Experimental design

Overall, this study highlights the importance of considering the impact of research methods on birds, particularly nestlings. Many ecological studies involve handling, and this research sheds light on how such handling can be a chronic stressor. Understanding how handling affects the HPA axis allows researchers to minimize stress or account for its effects in data analysis. The study offers a detailed and rigorous description of its methods, demonstrating a well-conducted experiment with interesting results. However, some limitations warrant consideration. There is a lot of scientific value in a study on the wild population of birds like yours and I would highlight that aspect of your study in the ‘Background’ section of the abstract. The scientific novelty stemming from this paper is not clear and it would be the Author's responsibility to make sure that their effort adds to previous ones in an important and creative manner. A key limitation of this study is that it examines only the effects of repeated handling on stress. However, most research likely involves additional manipulations like blood sampling or feather collection, which could contribute to cumulative stress in nestlings. The impact of these additional procedures on stress levels remains unexamined. Moreover, corticosterone provides valuable data, but it might not capture the full stress response picture. Future studies could benefit from including other physiological markers or behavioural observations.

Validity of the findings

The study focuses only on pre-fledging body mass and doesn't explore the consequences of the altered HPA axis on survival until fledgling or immune function. The study provides a rather narrow picture and does not allow us to draw any conclusions about how repeated sampling disturbances can affect offspring fitness-related traits.

---

## Round 0.2 · Minor Revisions

The reviewers have now returned with their comments. They both agree that the paper gained in quality and clarity. One last request from one of the reviewers is to add a brief mention (in Discussion and/or Intro) relating the outcomes of the study to a broader taxonomic context. I agree that birds are not the only animals heavily depending on the HPA axis, and pointing this out will be a valuable link between the present study and broader literature.

It is a small change and so I expect the paper to only go through my reading, without being sent out for another review.

·

Basic reporting

The revisions have further enhanced readability and clarity. I appreciate your incorporation of the suggested reference on glucocorticoid influences on memory. The updated introduction effectively contextualizes the study within the broader field, highlighting its relevance and novelty.The addition of pre-fledging tarsus length comparisons between treatment and control nestlings adds depth to the results. I also appreciate the provision of raw data and details on the corticosterone assay recovery range, which enhances transparency. The revised manuscript remains focused and self-contained, presenting results that are directly relevant to the hypotheses and research objectives.

Experimental design

This study represents original research that falls well within the journal's aims and scope. By examining the effects of chronic handling stress on the development of the HPA axis in nestling birds, it contributes novel findings to the field of avian physiological ecology. The revisions to the abstract and introduction further clarify the scientific novelty of the study, emphasizing the ecological relevance and practical utility of the findings. I am satisfied with the discussion of why brood size, hatch order, and sex were included as random effects. Additionally, your explanation regarding the handling time and its minimal impact on parental provisioning rates strengthens the reliability of the experimental design.

Validity of the findings

The study effectively highlights how its findings build on existing literature and provides a foundation for meaningful replication. Your acknowledgment of the limitations, such as the lack of long-term indices is appropriate and well-integrated into the discussion. The statistical analyses are robust and appropriate for the dataset. The addition of pre-fledging tarsus length comparisons provides valuable supplementary insights, and the revisions ensure all underlying data are clearly presented and controlled. The conclusions are well-stated, directly tied to the research question, and appropriately limited to the supporting results. The new discussion on potential mitigation strategies for handling stress adds practical relevance to the findings. The methods are described in sufficient detail to enable replication by future researchers.

·

Basic reporting

The article meets the necessary standards, with clear, unambiguous, and technically precise English used throughout. The text aligns well with expectations for scientific writing. The article provides an adequate introduction and background, with relevant literature appropriately cited. The structure follows a clear, standard format.
While this study was conducted specifically on birds, its findings hold broader significance for researchers working with a wide range of animals. The stress responses observed in nestlings could offer valuable insights for all researchers who capture and handle wildlife, not only for avian species. By framing the research within this broader context, the authors can underscore its relevance to the entire field of animal handling in ecological studies.

Experimental design

The article presents original primary research that addresses a well-defined and meaningful research question. The authors have identified a knowledge gap and demonstrated how their work contributes to filling this gap. Methods described with sufficient information.

Validity of the findings

I appreciate the authors' thorough revisions, which have successfully addressed the study's limitations as highlighted. I am now satisfied with the results. However, as I mentioned, while the study was conducted specifically on birds, its findings hold broader significance for researchers working with a wide range of animals. I would suggest that the authors emphasize this broader relevance in the Discussion and consider highlighting it in the Introduction or Abstract. This will help underscore that the insights provided are valuable not only for those studying birds but for all researchers involved in the handling and capture of wildlife.

---

## Round 0.3 · accepted · Accept

Thank you for your prompt reply to the most recent minor revision request. I'm pleased to say that you have incorporated all recommended edits, the paper can now be formally accepted for publication in PeerJ.